# Transport mechanism of P4 ATPase phosphatidylcholine flippases

Lin Bai[1†]*, Qinglong You[2†], Bhawik K Jain[3†], H Diessel Duan[2], Amanda Kovach[2], Todd R Graham[3], Huilin Li[2]*

[1]Department of Biochemistry and Biophysics, School of Basic Medical Sciences, Peking University, Beijing, China; [2]Department of Structural Biology, Van Andel Institute, Grand Rapids, United States; [3]Department of Biological Sciences, Vanderbilt University, Nashville, United States

**Abstract** The P4 ATPases use ATP hydrolysis to transport large lipid substrates across lipid bilayers. The structures of the endosome- and Golgi-localized phosphatidylserine flippases—such as the yeast Drs2 and human ATP8A1—have recently been reported. However, a substrate-binding site on the cytosolic side has not been found, and the transport mechanisms of P4 ATPases with other substrates are unknown. Here, we report structures of the *S. cerevisiae* Dnf1–Lem3 and Dnf2–Lem3 complexes. We captured substrate phosphatidylcholine molecules on both the exoplasmic and cytosolic sides and found that they have similar structures. Unexpectedly, Lem3 contributes to substrate binding. The conformational transitions of these phosphatidylcholine transporters match those of the phosphatidylserine transporters, suggesting a conserved mechanism among P4 ATPases. Dnf1/Dnf2 have a unique P domain helix-turn-helix insertion that is important for function. Therefore, P4 ATPases may have retained an overall transport mechanism while evolving distinct features for different lipid substrates.

*For correspondence:
lbai@bjmu.edu.cn (LB);
Huilin.Li@vai.org (HL)

[†]These authors contributed equally to this work

Competing interests: The authors declare that no competing interests exist.

## Introduction

The large group of integral membrane proteins that mediate the transport of substrates across biological membranes by the formation of ATP-dependent phosphorylated intermediates are called P-type ATPases (*Axelsen and Palmgren, 1998*). They are widely expressed in prokaryotes, archaea, and eukaryotes, and they play essential roles in many cellular processes. P-type ATPases are phylogenetically divided into five subclasses (P1-P5), each have unique sequence motifs and transport different substrates (*Bublitz et al., 2011*; *Palmgren and Axelsen, 1998*). Most P4 ATPases function as a heterodimeric complex: a catalytic α-subunit containing 10 transmembrane α-helices (TMHs) and a regulatory β-subunit of the Cdc50 (cell division cycle 50) protein family containing two TMHs. The α-subunit of all P-type ATPases has a conserved architecture consisting of a transmembrane domain (TMD), an actuator domain (A-domain), a nucleotide-binding domain (N domain), and a phosphorylation domain (P-domain) (*Andersen et al., 2016*; *van der Mark et al., 2013*). The 'Post–Albers' model proposed that the substrate transport of P-type ATPases goes through a cyclic transition of E1–E1P–E2P–E2 states (*Post et al., 1972*). This model is supported by the structure and function studies of the most well-understood P2 ATPases (*Dyla et al., 2019*; *Toyoshima and Cornelius, 2013*).

P4 ATPases are involved in phospholipid translocation: they flip lipids from the extracellular side of the plasma membrane or from the lumenal side of internal organelles to the cytosolic side (*Tang et al., 1996*; *Zhou and Graham, 2009*). They are important for creating and maintaining lipid compositional asymmetry in cell membranes. Because a phospholipid substrate is large, the transport mechanism of P4 ATPases is substantially different from that of the cation-transporting P-type ATPases, such as the sarcoplasmic reticulum $Ca^{2+}$-ATPase (SERCA), the $Na^+$/$K^+$-ATPase, and the $H^+$-

ATPase. There are five P4-ATPases in *Saccharomyces cerevisiae*: Drs2, Neo1, Dnf1, Dnf2 and Dnf3, belonging to at least three distinct phylogenetic classes (*Palmgren et al., 2019*; *van der Mark et al., 2013*). Humans have 14 P4-ATPases (*Andersen et al., 2016*; *Best et al., 2019*; *van der Mark et al., 2013*). The P4 ATPases have different tissue-specific expression, subcellular localization, and substrate specificity. For example, yeast Drs2 transports phosphatidylserine (PS) and phosphatidyl-ethanolamine (PE) in the trans-Golgi network (TGN) and endosomes (*Natarajan et al., 2004*; *Zhou and Graham, 2009*), while the yeast Dnf1 and Dnf2 flip PE, phosphatidylcholine (PC), and glu-cosylceramide (GlcCer) in the plasma membrane (*Best et al., 2019*; *Pomorski et al., 2003*; *Roland et al., 2020*; *Roland et al., 2019*; *Stevens et al., 2008*).

The first structural and mechanistic insights into the P4 ATPases came from recent cryo-EM struc-tures of the *S. cerevisiae* flippase Drs2–Cdc50 in its auto-inhibited and active E2P states (*Bai et al., 2019a*; *Timcenko et al., 2019*), and of its human homolog, ATP8A1–CDC50a, in six transport states (E1, E1-ATP, E1P-ADP, E1P, E2P, and E2Pi-PL) (*Hiraizumi et al., 2019*). Drs2 and ATP8A1 belong to the same subfamily of P4 ATPases, and both transport the same substrates, PS and PE (*Palmgren et al., 2019*; *Paterson et al., 2006*; *van der Mark et al., 2013*; *Zhou and Graham, 2009*). The structural studies revealed an allosteric domain movement during lipid translocation and identified a lipid-binding site at the exoplasmic side of membrane of this subfamily. Prior mutational studies had identified amino acids in this exoplasmic lipid-binding site that are important for sub-strate recognition and had implicated a second substrate-binding site on the cytosolic side of the membrane (*Baldridge and Graham, 2012*; *Baldridge et al., 2013*; *Baldridge et al., 2013*; *Roland et al., 2019*).

However, many important questions remain regarding the lipid-flipping mechanism(s) of the P4 ATPases. Chief among them are how substrate specificity is achieved by different P4 ATPases and where the lipid binds on the cytoplasmic side of the membrane. Addressing these questions requires a detailed structural and functional comparison among P4 ATPases that transport different sub-strates. Toward this goal, we have now studied the two *S. cerevisiae* PC flippases P4 ATPases, the Dnf1–Lem3 and Dnf2–Lem3 complexes. Dnf1 and Dnf2 share the same β-subunit, Lem3, and both are also capable of transporting PE and GlcCer (*Jain et al., 2020*). Their human homologs, ATP10A and ATP10D, are implicated in diabetes, obesity, myocardial infarction, atherosclerosis, and Parkin-son's disease (*Folmer et al., 2009*; *BELNEU consortium et al., 2020*; *van der Mark et al., 2013*). Here, we report the cryo-EM structures of Dnf1–Lem3 in three different states (E1, E1P-ADP, and E2P) and Dnf2–Lem3 in five different states (E1, E1-ATP, E1P-ADP, E2P, and E2P-transition). We fur-ther performed structure-guided mutagenesis and in vivo substrate-transport activity assays. By com-paring these PC flippase structures with the PS flippase structures previously reported, we found new phospholipid-binding sites at the cytoplasmic side and unique structural features during the ATP-dependent transport cycle in Dnf1–Lem3 and Dnf2–Lem3. Our comprehensive structural and functional study improves the mechanistic understanding of the essential P4 ATPases.

## Results and discussion

### Overall Dnf1 structure

Endogenous Dnf1–Lem3 and Dnf2–Lem3 complexes were purified from baker's yeast using an anti-FLAG affinity column followed by size-exclusion chromatography, and the protein particles had detailed features in the 2D averages of the cryo-EM images (*Figure 1A–B*, *Figure 1—figure supple-ment 1*, Materials and methods). We used the detergent dodecyl maltoside (DDM) to solubilize these flippases and lauryl maltose neopentyl glycol (LMNG) and cholesteryl hydrogen succinate (CHS) to stabilize the membrane protein complexes. To ascertain the activity of the purified proteins, we measured the ATPase activity of both Dnf1–Lem3 and Dnf2–Lem3 in the presence of either a known substrate GlcCer or a non-substrate sphingomyelin at 0.1 mM concentration. We found only the substrate GlcCer but not the non-substrate SM stimulated the ATPase activity of the enzymes, and the stimulation is concentration dependent (*Figure 1C–D*). Next, we performed single-particle cryo-EM on purified samples in apo form (E1 state, *Figure 1B*) and on samples incubated with AMPPCP (E1-ATP state), AlF4⁻-ADP (E1P-ADP and E2P-transition states), or BeF3⁻ (E2P state). The substrate-transport states were assigned based on the established nomenclature in the P-type ATPase field (*Bai et al., 2019a*; *Hiraizumi et al., 2019*; *Timcenko et al., 2019*). We obtained cryo-

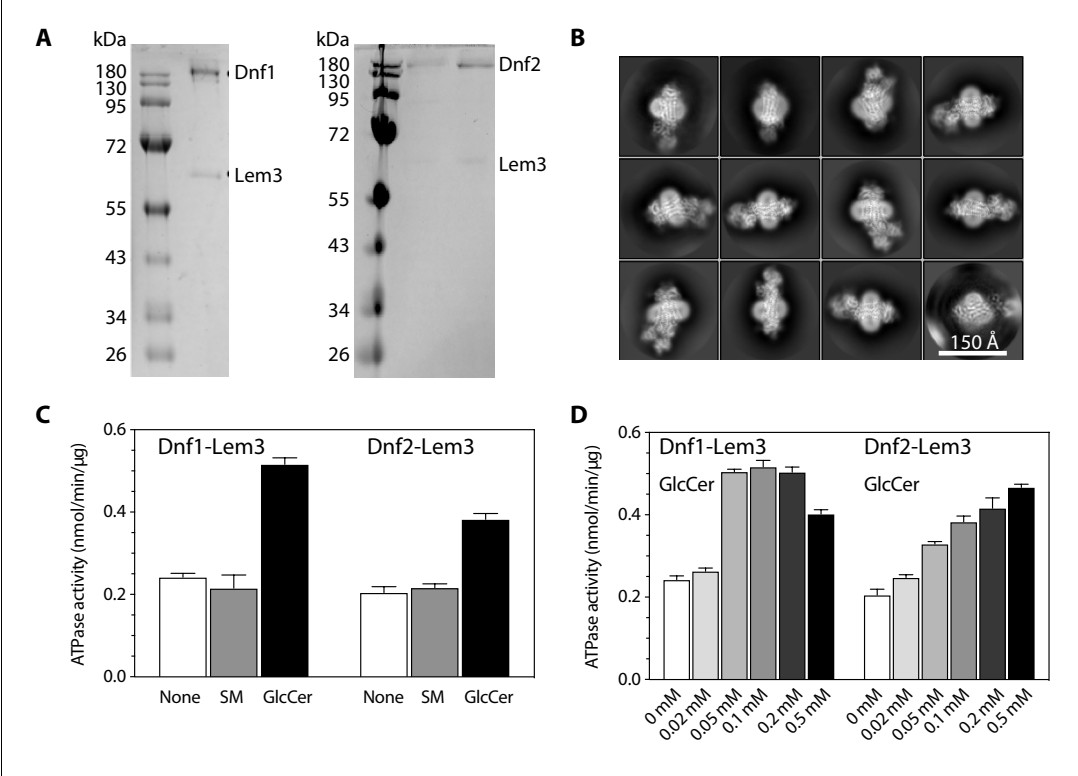

**Figure 1.** Purification and characterization of the detergent-purified flippases. (**A**) SDS-PAGE gel of the purified Dnf1-Lem3 (left) and Dnf2-Lem3 (right). (**B**) 2D class averages of cryo-EM images of Dnf1-Lem3 in the E2P state. (**C**) ATP hydrolysis activity of Dnf1-Lem3 and Dnf2-Lem3 without added substrate or with the addition of 0.1 mM sphingomyelin (non-substrate) or 0.1 mM GlcCer (substrate). (**D**) ATP hydrolysis activity of Dnf1-Lem3 and Dnf2-Lem2 at increasing GlcCer concentration. Data points in (**C–D**) represent the mean ± SD in triplicate.

The online version of this article includes the following figure supplement(s) for figure 1:

**Figure supplement 1.** Purification of the Dnf1–Lem3 and Dnf2–Lem3 complexes.

EM 3D maps of the *S. cerevisiae* Dnf1–Lem3 in the E1, E1P-ADP, and E2P states and Dnf2–Lem3 in the E1, E1-ATP, E1P-ADP, E2P-transition, and E2P states, at resolutions of 2.8 Å to 4.0 Å (*Figure 2—figure supplements 1–9*, *Supplementary file 1*). The high-resolution 3D map at 2.8 Å of the Dnf1–Lem3 in the E2P state enabled us to build the first atomic model of Dnf1–Lem3 (*Figure 2*, *Figure 2—figure supplement 1*). Except for a few short and disordered loops, almost the entire complex is ordered and well resolved in this model. Interestingly, many extra densities were observed in the transmembrane region of Dnf1 in the E2P state (*Figure 2A*). Some clearly belonged to the detergent CHS that was added during purification, while others located in the substrate-transport path resembled phospholipid, and they were so assigned, as will be described below.

We then used the 2.8 Å structure to help build atomic models for the seven remaining 3D maps of Dnf1p–Lem3p and Dnf2p–Lem3p, which are at slightly lower resolutions. All these models were refined to good statistics (*Supplementary file 1*) and fit well with the 3D maps (*Figure 2—figure supplements 2*, *4*, *6*, *8* and *9*). The modeled nucleotides and lipids all had clear densities. By comparing the 3D maps of Dnf1–Lem3 and Dnf2–Lem3 in the E1, E1P-ADP, and E2P states, which were the three states observed for both flippases, we found that the structures were superimposable in all three states (*Figure 2—figure supplement 10*). This comparison suggests that Dnf1 and Dnf2 have highly similar structures and go through virtually the same functional states during their respective substrate-transport cycles. This finding was not surprising, because these two flippases share a high sequence identity of about 69%, localize to the same membrane, transport the same substrates, and appear to have redundant functions in yeast (*Pomorski et al., 2003*). Because of their apparently interchangeable structural feature, we will discuss these two flippases together in the following sections.

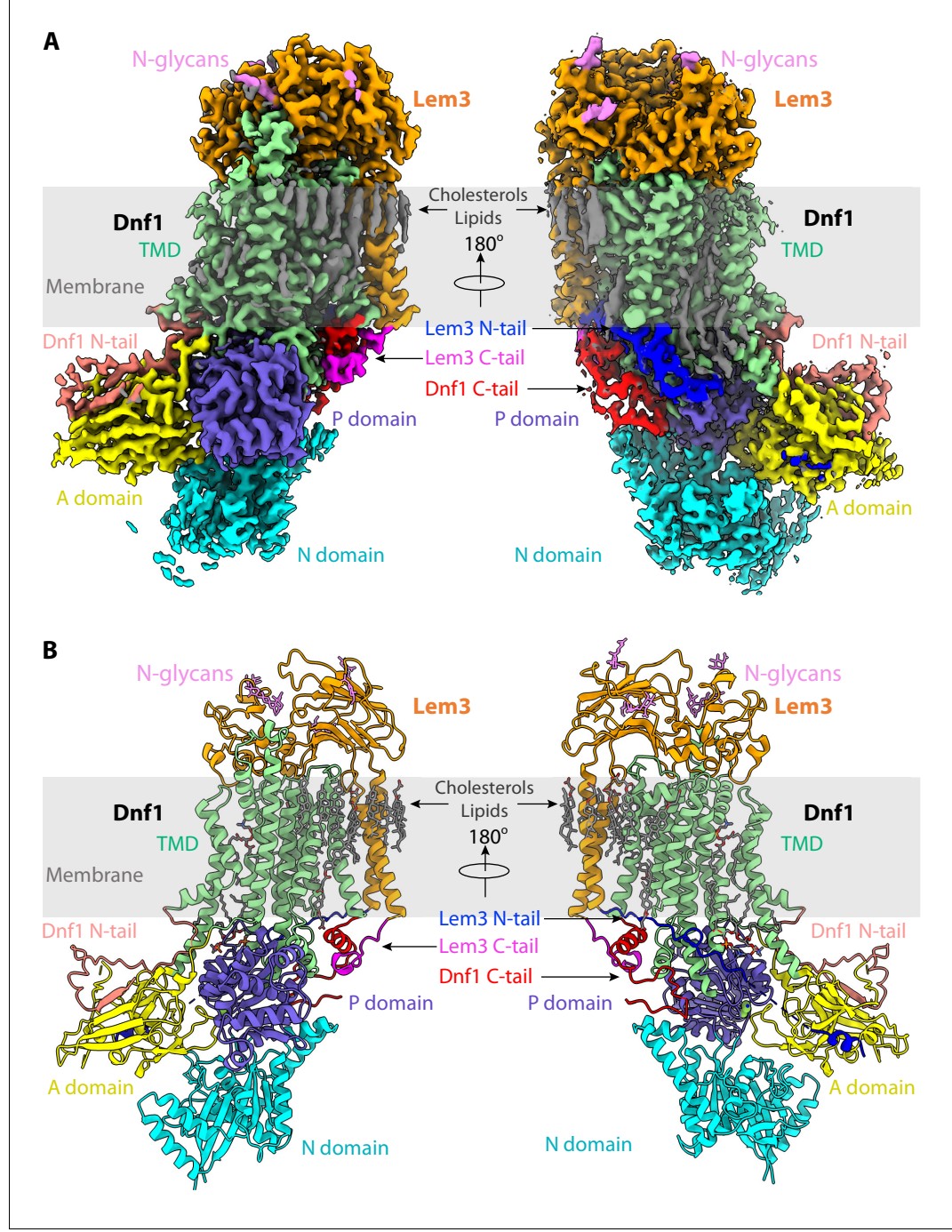

**Figure 2.** Structure of the *S. cerevisiae* class-3 lipid flippase Dnf1–Lem3 in the E2P state. (**A**) Cryo-EM 3D map of Dnf1–Lem3 in the E2P, in front (left) and back (right) views. The major domains and motifs are labeled in different colors. (**B**) Atomic model of Dnf1–Lem3 in the E2P state in cartoon and colored as in panel A. The detergent and phospholipid molecules and the covalently linked glycans on Lem3 are shown as sticks.

The online version of this article includes the following figure supplement(s) for figure 2:

**Figure supplement 1.** Cryo-EM data processing and resolution estimation of Dnf1–Lem3 in the E2P state.

**Figure supplement 2.** Selected regions in the 3D map of Dnf1–Lem3 in the E2P state, superimposed on the atomic model.

**Figure supplement 3.** Cryo-EM data processing and validation of the Dnf1–Lem3 in the E1 and E1P-ADP states.

**Figure supplement 4.** Transmembrane helices in the 3D map of Dnf1–Lem3 in the apo E1 state (**A**) and E1P-ADP state (**B**), superimposed on the atomic model.

*Figure 2 continued on next page*

*Figure 2 continued*

**Figure supplement 5.** Cryo-EM data processing and resolution estimation of the Dnf2–Lem3 in the E1 and E1-ATP states.

**Figure supplement 6.** Transmembrane helices in the 3D map of Dnf2–Lem3 in the apo E1 state (**A**) and E1-ATP state (**B**), superimposed on the atomic model.

**Figure supplement 7.** Cryo-EM data processing and validation of the Dnf2–Lem3 in the E2P-transition state and in the E1P-ADP state.

**Figure supplement 8.** Transmembrane helices in the 3D map of Dnf2–Lem3 in the E2P-transition state (**A**) and E1-ADP state (**B**), superimposed on the atomic model.

**Figure supplement 9.** Cryo-EM data processing and validation of the Dnf2–Lem3 in the E2P state.

**Figure supplement 10.** Structural alignments between the Dnf1 and Dnf2 in their corresponding E1 (apo) (**A**), the E1P-ADP (**B**), and the E2P states (**C**) show a high similarity.

**Figure supplement 11.** The N-terminal peptide of Lem3p is partially ordered in the 3D map of Dnf1–Lem3 in the E2P state.

We first examined the Dnf1 structure in the E2P state because of the high-resolution and completeness of the structure. As expected of any P4 ATPase, Dnf1 has a 10-TMH TMD (TMH1-10); a cytosolic A domain inserted between TMH2 and TMH3; and a cytosolic P domain and a cytosolic N domain, both inserted between TMH4 and TMH5 (*Figure 2*). The amino terminal peptide preceding TMH1 (N-tail) of Dnf1 is 204 residues long and is largely disordered, except for the last 39-residue segment (Glu-166 to Leu-204), which is stabilized by binding to the regulatory A domain. The carboxyl terminal peptide (C-tail) of Dnf1 following the 10th TMH is 157 residues long (Pro-1405 to Asn-1571) and is also largely disordered except for the TMD-proximal 35-residue segment (Pro-1405 to Lys-1439).

Interestingly, this ordered Dnf1 C-tail segment contains a short α-helix, the equivalent of which was reported to function as a switch that converts the Drs2 flippase from its auto-inhibitory state to the active state in a unique PI4P-binding-dependent manner (*Bai et al., 2019a*). However, the Dnf1 helix is stabilized by the C-tail of Lem3 in a conformation corresponding to the active state of Drs2, suggesting that Dnf1 and Dnf2 do not require binding to a phosphoinositide at this site for activation. The Dnf1 C-tail interacts extensively with the cytoplasmic domains of Dnf1 in regulating the substrate-transport cycle, as described below. Dnf1 and Dnf2 activity is regulated by two flippase kinases Fpk1 and Fpk2. Mutations in Fpk1 and Fpk2 do not affect the subcellular location of Dnf1 and Dnf2, but these enzymes have almost no activity in a *fpk1,2Δ* strain (*Nakano et al., 2008*; *Roelants et al., 2010*). We found that all 17 known phosphorylation sites (P32660 in the UniProt Database [*UniProt Consortium, 2019*]) are located in disordered loop regions in the Dnf1 structure: the N-tail, a loop between Thr-290 and Ala-378, and the C-tail. These observations suggest that the PC flippases are autoinhibited and phosphorylation, not PI4P binding as in the yeast PS flippases, relieves the inhibition.

## Lem3 has unique and ordered N- and C-terminal regions

Similar to the yeast Cdc50 and human CDC50a, Lem3 contains two TMHs flanking a large lumenal domain. These TMHs are packed against the last TMH of Dnf1, TMH10 (*Figure 2*). The Lem3 lumenal domain contains two disulfide bonds (Cys-110 with Cys-159 and Cys-216 with Cys-231) and four N-glycans at Asn-240, Asn-256, Asn-298, and Asn-332. To our surprise, both the N-terminal and the C-terminal regions of Lem3 are stable, in contrast to the yeast Cdc50 and human CDC50a, in which these regions are disordered (*Figure 2A–B*). The N-terminal peptide preceding Asn-50 folds into a two-turn α-helix and binds the Dnf1 A domain (*Figure 2—figure supplement 11A–B*). Due to a partial flexibility, the exact sequence of the Lem3 N-tail α-helix was not resolved. However, we found its binding interface in the A domain is highly positively charged, featuring Lys-392, Lys-395, Arg-408, Arg-457, Lys-460, and Lys-516 (*Figure 2—figure supplement 11C*), and the Lem3 N-tail segment between Glu-28 to Glu-42 is highly negative, containing 11 Asp and Glu residues (*Figure 2—figure supplement 11D*). Therefore, the short N-tail α-helix of Lem3p is most likely from this negatively charged region. Because the A domain moves significantly during the substrate-transport cycle of P4 ATPase, this unique, A domain-interacting, and highly negative Lem3 N-tail α-helix likely regulates lipid transport activity of Dnf1 and Dnf2, as described below.

## Phospholipids bind at the exoplasmic and cytoplasmic sites of Dnf1

Although the exact substrates a P4 ATPase transport differ, the substrate translocation path appears to have been conserved in all P4 ATPase studied so far (*Andersen et al., 2016*; *Huang et al., 2020*). Like the Drs2 and ATP8A1 flippases, we found that the Dnf1 substrate translocation path is composed of TMH1-4 and TMH6 (*Figure 3A–D*, *Figure 3—figure supplement 1*). This substrate path is consistent with biochemical studies demonstrating that TMH1-6 are responsible for phospholipid transport, with TM7-10 having a supporting role (*Baldridge and Graham, 2013*; *Vestergaard et al., 2014*). In the E2P-state Dnf1 3D map, we observed three phospholipid-like densities in the substrate-transport path, and accordingly, we modeled three phospholipid molecules (PL1-PL3) in that map (*Figures 2A* and *3A*). These phospholipids stack against Dnf1 TMH2, TMH4, and TMH6. Because no lipids were added during purification, these phospholipids are potentially endogenous substrates that were co-purified with the Dnf1–Lem3 complex. This is not unusual, because previous studies have shown that substrate lipids bound in the substrate pockets can be co-purified with the enzyme complexes (*Bai et al., 2019b*; *Bloch et al., 2020*; *Hiraizumi et al., 2019*). Interestingly, the three phospholipids are out of register with respect to the lipid bilayer by approximately 10 Å: PL1 is moved down about 10 Å into the middle of the bilayer relative to lipids in the outer leaflet, and PL2 and PL3 are moved 10 Å into the cytosolic space relative to lipids in the inner leaflet of the membrane.

## Functional studies reveal that PL1 and PL2 are in substrate-binding sites

Based on sequence conservation of the three phospholipid-binding pockets, we expect that PL1 and PL2—but not PL3—are located in the substrate path and are likely physiologically relevant. To test this hypothesis, we carried out systematic, structure-based mutagenesis and functional assays (*Figure 4A–E*, *Supplementary file 3*). We first demonstrated that all mutations we introduced in Dnf1 and Lem3 did not significantly affect the expression levels of the flippase (*Figure 4—figure supplement 1*). PL1 is bound in an open cleft of Dnf1 on the exoplasmic side of the plasma membrane, and the phosphate group of PL1 interacts with several conserved residues, including Gln-610, Ser-611, and Asn-1226 (*Figure 3A–B*). We expressed the Q610A, S611A, 609YQS611-FSN, and N1226A Dnf1 mutants on a *dnf1Δ dnf2Δ* cell background and measured their PC, PE, and GlcCer uptake abilities. We found that Q610A and the triple mutation 609YQS611-FSN eliminated the GlcCer and PE, but not the PC, transport activity (*Figure 4A*). This observation was consistent with a previous finding in which mutations of the conserved tripeptide motif 654YQS656 of Dnf2—the equivalent of 609YQS611 in Dnf1—caused substantial and specific loss of GlcCer transport (*Roland et al., 2020*; *Roland et al., 2019*).

It is unclear why the S611A mutation led to increased transport activity for all three substrates (*Figure 4A*). In contrast, the N1226A mutation completely halted transport of all three substrates, even though the mutant protein localized to the plasma membrane comparably to WT Dnf1 (*Figure 4D*). The ability of Dnf1-N1226A to exit the ER indicated it was folded properly and was able to interact with Lem3, but consistent with the loss of transport activity, Dnf1-N1226A failed to support the viability of a flippase-deficient yeast strain, whereas Dnf1 variants that retained transport activity fully supported viability (*Figure 4—figure supplement 2*). The GA motif (230-231) in TMH1B is at the membrane-exoplasmic interface and is next to the PL1-binding site (*Figure 3D*). A recent study has shown that the GA motif is crucial for GlcCer transport by Dnf2 but has little influence on PC and PE transport (*Roland et al., 2019*). Furthermore, we found that the PL1-binding site superimposed with the substrate-binding site of ATP8A1, based on structural alignment (*Figure 3—figure supplement 1*). Taken together, we conclude that the PL1-binding site is the entrance of the substrate-transporting path in Dnf1 and Dnf2.

Unlike PL1, both PL2 and PL3 bind Dnf1 on the cytosolic side of the lipid bilayer. The head group of PL2 is in a positively charged groove and is stabilized by several conserved residues, including Arg-264, Tyr-633, Thr-648, and Trp-652 of Dnf1, and Arg-51 of Lem3. Because PL2 is in the putative substrate-transport path, we suggest that PL2 is at the substrate exit site of Dnf1. We generated a series of point mutations in Dnf1, including R264A, Y633A, T648A, W652A, and W652S. We expressed these mutations in *dnf1Δ dnf2Δ*, *lem3Δ*, or *lem3Δ dnf2Δ* cells and found that W652A and W652S significantly disrupted transport of all substrates; that R264A enhanced PC, PE, and GlcCer

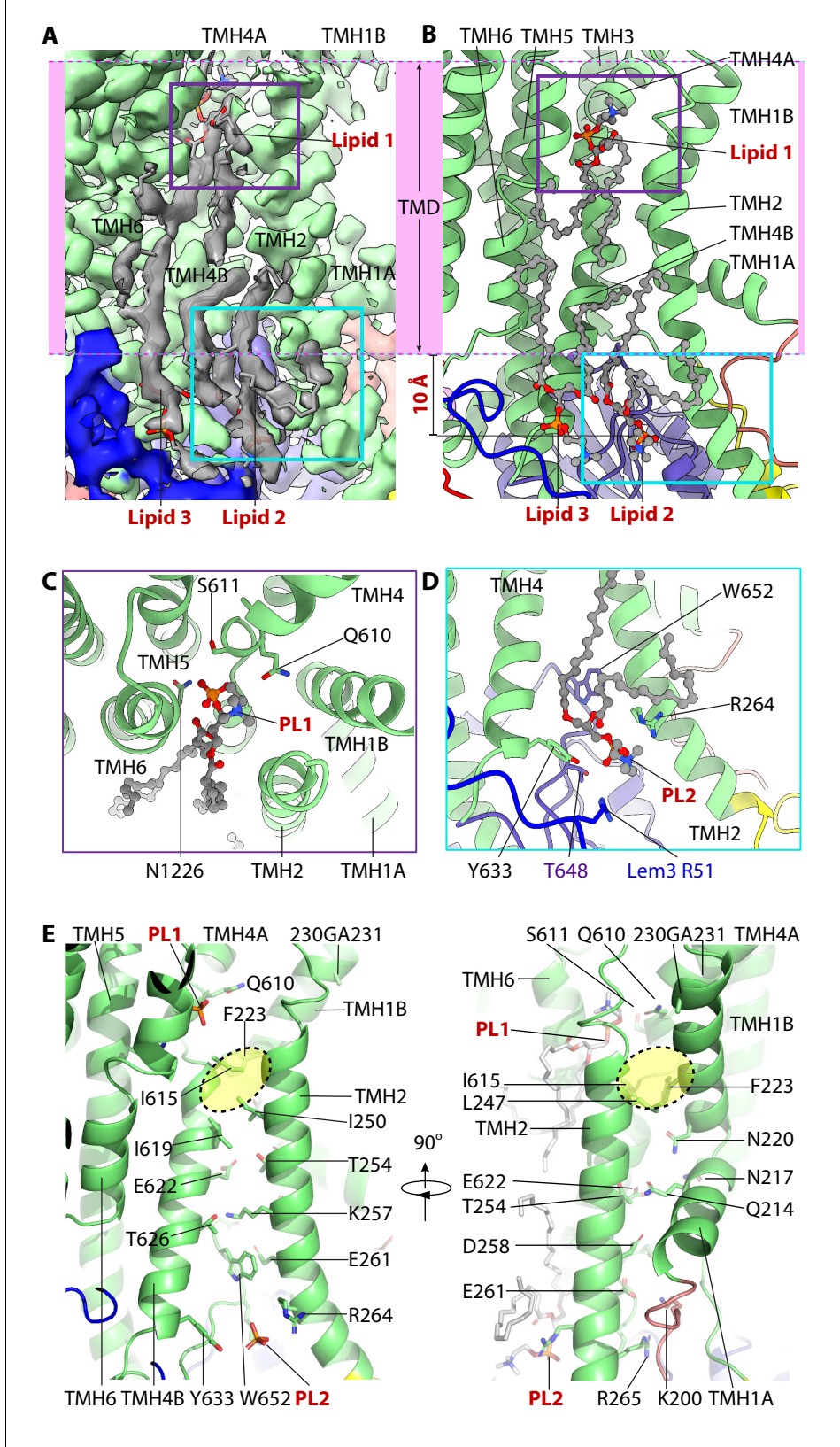

**Figure 3.** The lipid-binding sites and the putative substrate-transport path of Dnf1–Lem3. (**A**) Cryo-EM 3D of the region around the putative substrate-transporting path. The transmembrane region is marked by the pink band. Three lipids densities are shown in gray. PL1 in the exoplasmic side and PL2 in the cytosolic side of the membrane

*Figure 3 continued on next page*

*Figure 3 continued*

are highlighted by a purple and a cyan box, respectively. (**B**) Cartoon view of the lipid-binding region in the Dnf1–Lem3 structure. The purple box in panel A and the cyan box in panel B are enlarged in panels C and D, respectively. Note that PL1 is about 10 Å below the lipid bilayer at the exoplasmic side, and PL2 and PL3 ingress into the cytosolic region by 10 Å. (**C**) Close-up and top view of PL1-binding site. (**D**) Close-up view of PL2-binding site. (**E**) Putative substrate-transport path of Dnf1p, with the residues lining the path and bound PL1 and PL2 shown as sticks. The hydrophobic gate in the lipid transport path is highlighted by the yellow oval. PL3 is removed in the left panel for clarity.

The online version of this article includes the following figure supplement(s) for figure 3:

**Figure supplement 1.** Structural comparison between Dnf1 (color) and the ATP8A1 (gray) in the E2P state.

transport; and that Y633A and T648A had no significant effects on the transport activities (*Figure 4A*). The elimination of Dnf1 transport activity by the W652 mutations was consistent with this residue's strategic location between the PL2-binding substrate exit site and the PL1-binding substrate entry site. However, this mutation caused the retention of Dnf1 in the ER, possibly because it disrupts folding of the protein (*Figure 4E*). The increased activity of Dnf1 R264A suggests that this charged residue may trap the substrates and slow their release into the cytosolic leaflet. This observation supports our hypothesis that PL2 is a substrate lipid. Because the PL3 site is located right next to the PL2 site, and the PL3 site is composed of nonconserved residues (except for F631), we suggest that the PL3 site plays an accessory role, perhaps by facilitating substrate release from the PL2 site.

## Lem3 unexpectedly contributes to substrate binding on the cytosolic side

Because the Lem3 Arg-51 directly interacts with PL2, we generated a Lem3 R51A mutant and found that the substitution changed the substrate specificity of both Dnf1 and Dnf2, with a 1.5- to 2-fold increase in GlcCer transport activity of Dnf2–Lem3 and Dnf1–Lem3, but no change in PC or PE activity for either complex (*Figure 4B–C*). Because there is no phosphate in GlcCer, it is possible that R51A increases ability of GlcCer to compete with endogenous lipids for this site. As expected, Lem3-R51A supported trafficking of the complex out of the ER (*Figure 4E*), and this result supports the role of Lem3 R51 in controlling substrate specificity and further supports the assignment of PL2 as a substrate lipid.

To examine the functional importance of the N-terminal region of Lem3, we generated a Lem3 variant missing the N-terminal peptide (Lem3[Δ2–49]) and performed a lipid uptake assay. We found that with Lem3(Δ2–49), both Dnf1 and Dnf2 had significantly reduced PC, PE, and GlcCer uptake ability (*Figure 4B–C*). We next examined the functional importance of the ordered C-terminal region of Lem3 (Gly-400 to Lys-414) that stabilizes Dnf1 in an active state. In the lipid uptake assays with Lem3(Δ400–414), both Dnf1 and Dnf2 had substantially decreased activity (*Figure 4B–C*). We then labeled Dnf1 with GFP and found that Lem3(Δ2–49) and Lem3(Δ400–414) abolished Lem3's ability to escort Dnf1 out of the ER (*Figure 4E*). Thus, neither Dnf1 or Dnf2 was able to reach to its functional destination in the plasma membrane with a N- or C-terminus truncated Lem3. Thus, these results support an unexpected role of Lem3 in substrate binding and specificity. No such role was found in the β-subunit (Cdc50 or CDC50) of the class-1 flippases Drs2 and ATP8A1 (*Bai et al., 2019a*; *Hiraizumi et al., 2019*; *Timcenko et al., 2019*).

During preparation of this manuscript, the cryo-EM structure of Dnf1–Cdc50 of a thermophilic fungus, *Chaetomium thermophilum*, was reported in the E1-ATP and E2P states at 3.4–3.5 Å resolution (*He et al., 2020*). That Dnf1 structure contained two lipids, but no evidence was provided that they were substrate lipids. Many membrane proteins, including SERCA and Na$^+$/K$^+$ ATPases (*Kanai et al., 2013*; *Toyoshima et al., 2013*), have specifically bound lipid molecules in their structures that are presumably not substrate lipids. Because we have now functionally defined that the PL1 and PL2 in our *S. cerevisiae* Dnf1–Lem3 and Dnf2–Lem3 structures are located at the physiological substrate entry and exit, the observation that the two lipids in *C. thermophilum* Dnf1 are equivalent to PL1 and PL2 in our structure indicates that in both species, the PC flippases share a similar substrate path.

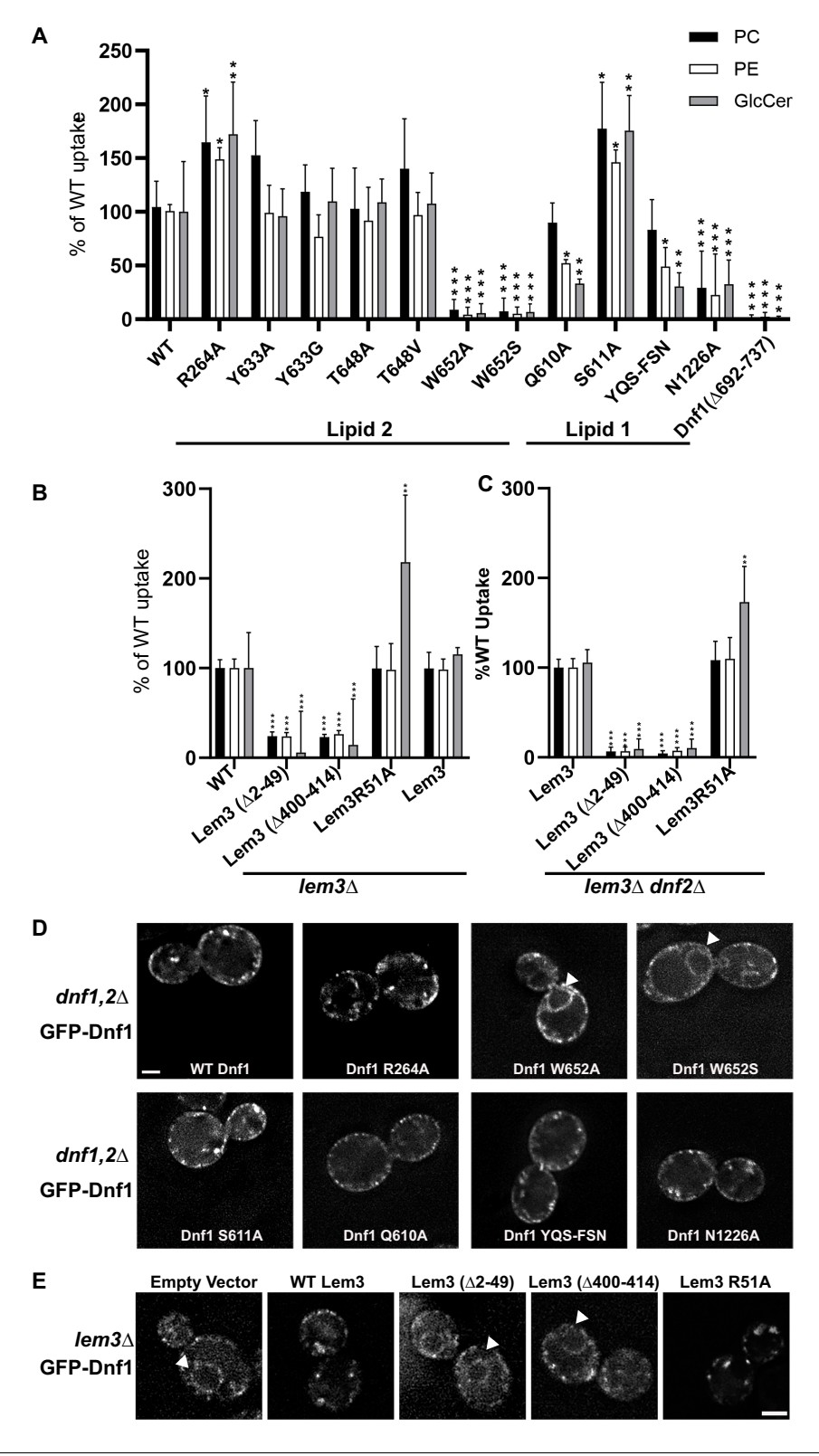

**Figure 4.** Influence of lipid=-binding site mutations on Dnf1 transport activity and localization. (**A**) NBD-lipid uptake catalyzed by Dnf1 variants expressed on a *dnf1,2Δ* strain background. (**B**) NBD-lipid uptake catalyzed by Lem3 variants expressed on a *lem3Δ* background, in which lipid transport is mediated primarily by Dnf2–Lem3. (**C**) NBD-lipid uptake catalyzed by Lem3 variants expressed on a *lem3Δ dnf2Δ* background to assay the influence of Lem3 mutations solely on Dnf1 activity. In (**A–C**), cells were incubated with NBD-lipids for 30 min and uptake was measured by flow cytometry. The
*Figure 4 continued on next page*

*Figure 4 continued*

value represents percentage uptake of NBD-lipid in comparison to WT Dnf1 (**A**) or WT Lem3 (**B–C**) for each substrate lipid. The variance was assessed with one-way ANOVAs, and comparison to WT was calculated with Tukey's post hoc analysis. $n \geq 9, \pm$ SD, * indicates $p<0.05$, **$p<0.01$, ***$p<0.001$. (**D**) Localization of GFP-tagged Dnf1 variants expressed in *dnf1,2Δ* cells. (**E**) WT Dnf1-GFP co-expressed with Lem3 variants. Arrowheads indicate nuclear envelope/endoplasmic reticulum location. Scale bar = 2 μm (*n* = 40).

The online version of this article includes the following figure supplement(s) for figure 4:

**Figure supplement 1.** Expression levels of Dnf1 variants.

**Figure supplement 2.** Ability of Dnf1 variants to support viability of flippase-deficient cells.

We conclude that the substrate-transport pathway is primarily located between TMH2 and TMH4 (*Figure 3E*). This pathway is nearly perpendicular to the membrane plane and is open to the lipid bilayer. Interestingly, this pathway is hydrophilic at both ends, the entry and the exit sites, but is highly hydrophobic in the middle gating region (*Figure 3B–E*). In the Dnf1–Lem3 structure, the exoplasmic substrate entry site is lined by Gln-610, Ser-611, and Asn-1226; the cytosolic substrate exit site is surrounded by Dnf1 residues Gln-214, Asn-217, Asn-220, Thr-254, Lys-257, Asp-258, Glu-261, Arg-264, Arg-265, Glu-622, and Thr-626, along with Lem3 Arg-51, and the middle hydrophobic region is composed of the conserved Phe-223, Leu-247, Ile-250, Ile-615, and Ile-619 (*Figure 3B–E*). The physiochemical property of the transport pathway is compatible with the so-called 'credit card' mechanism, in which the hydrophilic or charged headgroups of the lipid substrates simply slide through the groove while allowing the substrate acyl chains to flip in the favorable membrane environment. It is likely that during transport, the lipid head group inserts into and make direct interactions with the hydrophilic residues between TMH2 and TMH4.

## ATP-dependent transport cycle

By cryo-EM analyses of samples incubated with or without the ATP analogs AMPPCP and ADP-AlF$_4^-$ or the phosphate analogs AlF$_4^-$ and BeF$_3^-$, we obtained a total of eight structures of the class-3 flippases. There were Dnf1–Lem3 in the E1 (apo), E1P-ADP (AlF$_4^-$-ADP), and E2P (BeF$_3^-$) states; and Dnf2–Lem3 in the E1 (apo), E1-ATP (AMPPCP), E1P-ADP (ADP-AlF$_4^-$), E2P-transition (AlF$_4^-$), and E2P (BeF$_3^-$) states. Comparing these structures revealed how ATP binding and hydrolysis, ADP release, and enzyme phosphorylation produce conformational changes in Dnf1 and Dnf2 to drive the Post–Albers substrate-transport cycle (*Figure 5A*).

We found that a major part of the TMD region (that is, TMH3-10) and the P domain of Dnf1 or Dnf2, and the most of Lem3 molecule except for its N-tail, were well aligned, suggesting that these regions are largely stationary during the substrate-transport cycle (*Figure 5A*, *Figure 5—figure supplement 1A–B*). However, TMH1-2 and the A and N domains of Dnf1 and Dnf2 moved significantly; these movements must be coupled with substrate transport. In the E1 state, the A domain moves toward the membrane bilayer and becomes separated from the N domain (*Figure 5A*). Due to the lack of interaction, the A and N domains are flexible and are visible in the 3D maps only at a low display threshold (*Figure 5—figure supplement 2*).

The binding of ATP in the E1-ATP state triggers the N domain to move toward and interact with the A domain, thereby stabilizing both domains. However, the A domain must be only partially stabilized and retain some flexibility, because its density is still weak. With the ensuing phosphoryl transfer, the flippases transition from the E1-ATP state to the E1P-ADP state. This transition does not trigger large-scale conformational changes, because the structures of Dnf2–Lem3 in the E1-ATP and E1P-ADP are almost the same except for their respective bound nucleotide (*Figure 5—figure supplement 1*). In the E1, E1-ATP, and E1P-ADP states, TMH1-2 have a small degree of flexibility, but they do not move much (*Figure 5B*). Transitioning from the E1P-ADP state to the E2P state, the release of ADP triggers the N domain to rotate 40° away from the A domain by pivoting on the N domain and P domain interface. The conserved DGET motif of the A domain now occupies the phosphorylation site and stabilizes the A domain. During this transition, TMH1 and TMH2 tilt to enlarge the exoplasmic substrate-binding site for the admission of PL1 and at the same time to narrow the cytosolic substrate site, thereby stabilizing the bound PL2 (*Figure 5B*).

During the subsequent phosphate (Pi) transition, when the enzyme changes from the E2P state to the E2P-transition state, the enzyme structure largely holds still; the 3D maps of Dnf2–Lem3 in these

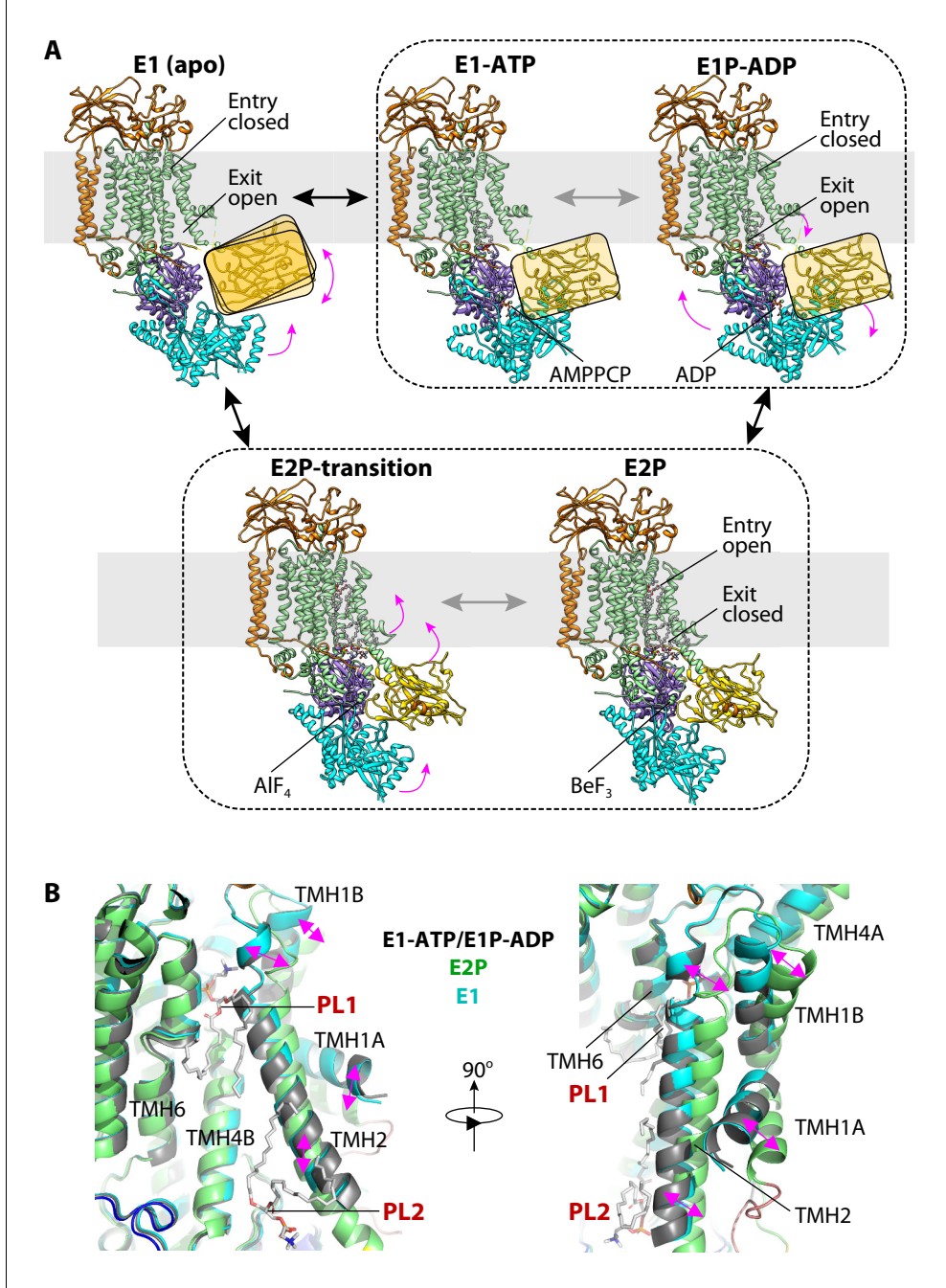

**Figure 5.** The ATP-dependent transport cycle of Dnf1–Lem3 and Dnf2–Lem3. (**A**) The five structures of Dnf1 and Dnf2 in the E1 (apo), E1-ATP (AMPPCP), E1P-ADP (AlF$_4^-$-ADP), E2P-transition (AlF$_4^-$), and E2P (BeF$_3^-$) states. The A domain in the E1, E1-ATP, and E1P-ADP states is flexible and is shown here as a ribbon in the yellow rectangle to indicate its position. The structures of the E1-ATP and E1P-ADP states within the black dashed box are almost identical. The structures of the E2P and E2P-transition states are similarly grouped for the same reason. Magenta arrows indicate the domain movement during the Post–Albers cycle in the order E1 to E1-ATP to E1P-ADP to E2P/E2P-transition. (**B**) Structural comparison of the substrate-binding sites in the E1, E1P-ADP, and E2P states. TMH1-2 in the E1 and E1P-ADP states tilt with respect to that in the E2P state; this movement closes the PL1 site and opens the PL2 site.

The online version of this article includes the following figure supplement(s) for figure 5:

**Figure supplement 1.** High structural similarity between Dnf1 and Dnf2 in the corresponding substrate-transport states.

*Figure 5 continued on next page*

Figure 5 continued

**Figure supplement 2.** The N and A domain densities are weak in the 3D maps of Dnf1–Lem3 in the E1 and E1P-ADP states, indicating their high flexibility.

two states are nearly identical (*Figure 5—figure supplement 1B*). Finally, the Dnf1 and Dnf2 switch back to the E1 state by auto-dephosphorylation. During this switch, the A and N domains become flexible again, and the TMH1-2 tilt back to narrow and close the exoplasmic substrate-binding site and open the cytosolic substrate-binding site. It is during this last transition that PL1 in the exoplasmic site is forced to flip and enter the cytosolic substrate site, replacing the resident PL2 there and leading to the release of PL2 from the flippases.

## Transport mechanisms of PC- and PS-specific P4 ATPases are similar

As described above, the major domain rearrangements in Dnf1 and Dnf2 are triggered by nucleotide binding (E1 to E1-ATP); nucleotide hydrolysis, enzyme phosphorylation, nucleotide release (E1P-ADP to E2P); and enzyme dephosphorylation (E2P to E1). The energy gain from ATP during these transitions is transduced from the phosphorylation site in the P domain to the A domain, which in turn causes the TMH1-2 to tilt and drive the flipping of the phospholipids. TMH3-10 of Dnf1 or Dnf2 and TMH1-2 of Lem3 largely remain stationary throughout the transport cycle. Combining the finding of two substrate-binding sites and the transport cycle, we propose that the lipid head group transports through the groove between TMH2 and TMH4, while lipid tails move accordingly on the hydrophobic surface of the TM2 and TM4. This is essentially the same as the previously described 'credit card model' (*Andersen et al., 2016*; *Roland and Graham, 2016*).

The previous cryo-EM study of the human ATP8A1–CDC50A flippase captured multiple transport states (*Hiraizumi et al., 2019*). Aligning our structures in this paper with those of ATP8A1–CDC50a reveal that the corresponding E1, E1-ATP, E1P-ADP, E1P, and E2P states are remarkably similar in both domain architecture and substrate-transport path (*Figure 6*). This results suggests a common, conserved transport pathway regardless of the lipid substrate being flipped. In a recent phylogenetic

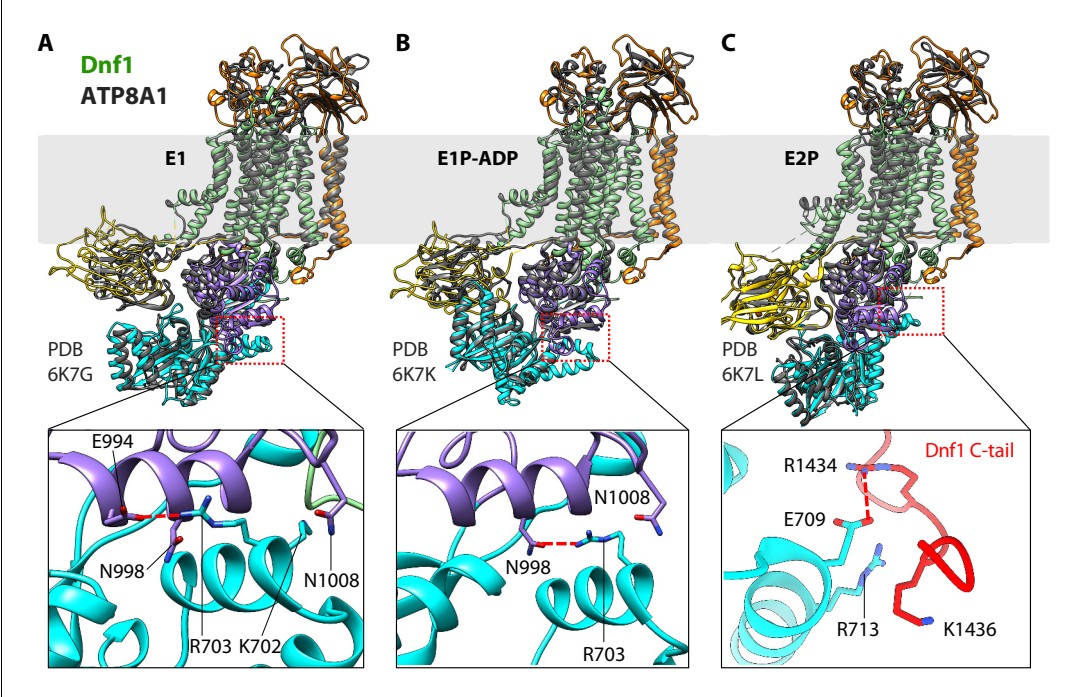

**Figure 6.** Structural comparison between Dnf1 (color) and the ATP8A1 (gray) in the E1 (A), E1P-ADP (B), and E2P (C) states. Bottom panels are close-up views of the boxed regions in the top panels. These panels highlight the unique helix-turn-helix motif in the N domain of Dnf1, which interacts with different regions in different states.

classification, P4-ATPases are divided into P4A, P4B and P4C subclasses (*Palmgren et al., 2019*). In this scheme, Dnf1 and 2 belong to the same P4A class as Drs2 and ATP8A1. The resemblance in structure and catalytic cycle between Dnf1/2 and Drs2/ATP8A1 supports this new classification scheme. In further agreement with this scheme, recent structural studies of the human ATP11C–CDC50A flippase, also a P4A member in the new scheme, have revealed a rather similar mechanism for lipid-flipping (*Nakanishi et al., 2020a*; *Nakanishi et al., 2020b*). While structural knowledge obtained thus far raises the possibility that all P4-ATPases employ a similar substrate-transport mechanism, more definitive support awaits further structure and functional analyses of P4-ATPases in the P4B and P4C classes.

Despite their overall similarity in substrate-transport cycle, notable structural differences exist among the flippases characterized so far. In PS flippases such as the yeast Drs2 and human ATP8A1, the C-terminal loop is auto-inhibitory; this loop, which contains the conserved GF/YAFS motif, extends over and wraps around the cytoplasmic catalytic domains to stabilize these enzymes in the E2P state. The C-terminal loop of Dnf1 and Dnf2 doesn't contain the GF/YAFS motif and is disordered in the E2P state. However, there is a unique helix-turn-helix motif (E692-S737) in the N domain of Dnf1 and Dnf2 that is absent in Drs2 and ATP8A1 (*Figure 6*). In the E1 state, the helix-turn-helix interacts with the P domain, specifically, its Lys-702 and Arg-703 interacting with the P domain Glu-994, Asn-998, and Asn-1008. Upon ATP hydrolysis and enzyme phosphorylation in the E1P-ADP state, this interaction is weakened because only Arg-703 of the motif now interacts with the P domain Asn-998 and Asn-1008. The release of ADP in the E2P state appears to shift the interaction of the Dnf1 and Dnf2 insertion from the P domain to the C-tail, with the helix-turn-helix Glu-709 and Arg-713 forming electrostatic interactions with the C-tail Arg-1434 and Lys-1436. These systematic changes suggest that the Dnf1 and Dnf2 insertion may play a regulatory role in the substrate-transport cycle. To examine this possibility, we deleted the motif in Dnf1 and performed an in vivo substrate uptake assay. The deletion strongly decreased transport of all substrates (*Figure 4A*). These results confirm a crucial role of the helix-turn-helix motif and reveal a unique aspect of the Dnf1/Dnf2 flippases over the otherwise conserved substrate-transport mechanism of the P4 ATPases.

In summary, our systematic structural and biochemical studies of the Dnf1–Lem3 and Dnf2–Lem3 complexes in five transport states provide the novel insight that the overall substrate-transport mechanism is similar to that of PS flippases and that it may be conserved among all classes of the P4 ATPases. Our studies also show a helix-turn-helix motif in the P domain that is unique to the class-3 enzymes and plays a crucial regulatory function during the substrate-transport cycle. Also unique to class-3 enzymes is the role of the beta-subunit Lem3 in substrate binding and specificity. Thus, this work advances our mechanistic understanding of the highly conserved and essential family of the P4 ATPases.

## Materials and methods

### Purification of the endogenous Dnf1p–Lem3p and Dnf2p–Lem3p complexes

Using a PCR-based genomic epitope-tagging method (*Funakoshi and Hochstrasser, 2009*), we tagged Dnf1p or Dnf2p with a C-terminal triple-FLAG in the yeast strain BY4741 (Horizon Discovery). Cells were grown in YPD medium for about 20 hr before harvest. Cells were resuspended in lysis buffer (20 mM Tris-HCl, pH 7.4, 0.2 M sorbitol, 50 mM potassium acetate, 2 mM EDTA, and 1 mM phenylmethylsulfonyl fluoride [PMSF]) and then lysed using a French press at 15,000 psi. The lysate was centrifuged at 10,000 × $g$ for 30 min at 4°C. The supernatant was collected and centrifuged at 100,000 × $g$ for 60 min at 4°C. The membrane pellet was collected and then resuspended in buffer A containing 10% glycerol, 20 mM Tris-HCl (pH 7.4), 1% DDM, 0.1% CHS, 0.5 M NaCl, 1 mM MgCl$_2$, 1 mM MnCl$_2$, 1 mM EDTA, and 1 mM PMSF. After incubation for 30 min at 4°C, the mixture was centrifuged for 30 min at 120,000 × $g$ to remove insoluble membrane. The supernatant was mixed with pre-washed anti-FLAG (M2) affinity gel at 4°C overnight with shaking. The affinity gel was then collected and washed three times in buffer B containing 20 mM HEPES, pH 7.4, 150 mM NaCl, 0.01% LMNG, 0.001% CHS, and 1 mM MgCl$_2$. The proteins were eluted with buffer B containing 0.15 mg/ml 3 × FLAG peptide and was further purified in a Superose 6 10/300 gel filtration column in buffer C containing 20 mM HEPES, pH 7.4, 150 mM NaCl, 0.003% LMNG, 0.0003% CHS and 1 mM MgCl$_2$.

Finally, the purified proteins were assessed by SDS-PAGE gel and concentrated for cryo-EM analysis. Approximate 20 μg Dnf1p–Lem3p and 40 μg Dnf2p–Lem3p can be purified from 18 L of cells.

## ATP hydrolysis assay of the detergent-purified lipid flippases

The ATPase activity assays were performed using BIOMOL Green Reagent (Enzo Life Sciences, Inc) to measure released inorganic phosphate. The lipids sphingomyelin and glucosylceramide were solubilized with 40 mM sodium cholate in 20 mM HEPES, pH 7.4, 150 mM NaCl. Each reaction contained 0.025 mg/mL protein, 0.003% LMNG, 0.0003% CHS, 20 mM HEPES, pH 7.4, 150 mM NaCl, 10 mM $MgCl_2$, 0.25 mM ATP. Reactions were carried out at 37°C for 15 min, and then terminated immediately by addition of the reagent. After incubation of the mixture for 20 min at room temperature, the absorbance at 620 nm was measured using a microplate reader (SpectraMax M2e). The phosphate concentration was determined by calibration with the phosphate standard (BML-KI102).

## Cryo-electron microscopy

To capture different states, the purified Dnf1p–Lem3p or Dnf2p–Lem3p was mixed with different solutions for 1 hr on ice: E1-ATP, 5 mM $MgCl_2$ and 2 mM AMPPCP; E1P-ADP and E2P-transition, 5 mM $MgCl_2$, 5 mM NaF, 1 mM $AlCl_3$, and 5 mM ADP; E2P, 5 mM $MgCl_2$, 10 mM NaF, and 2 mM $BeSO_4$. After an incubation, 2.5 μL aliquots of purified Dnf1p–Lem3p or Dnf2p–Lem3p at a concentration of about 1.5 mg/mL were placed on glow-discharged holey carbon grids (Quantifoil Au R2/2, 300 mesh) and were flash-frozen in liquid ethane using a FEI Vitrobot Mark IV. Cryo-EM data was collected automatically with SerialEM in a 300-kV FEI Titan Krios electron microscope with defocus values from −1.0 to −2.0 μm. The microscope was operated with a K3 direct detector at a nominal magnification of 130,000 × and a pixel size of 0.413 Å per pixel. The dose rate was eight electrons per $Å^2$ per s, and the total exposure time was 8 s.

## Cryo-EM image processing

Program MotionCorr 2.0 (*Zheng et al., 2017*) was used for motion correction, and CTFFIND 4.1 was used for calculating contrast transfer function parameters (*Mindell and Grigorieff, 2003*). All the remaining steps were performed using RELION 3 (*Zivanov et al., 2018*). The resolution of the map was estimated by the gold-standard Fourier shell correlation at a correlation cutoff value of 0.143.

For the Dnf1–Lem3 structure in the E2P state, we collected 4764 raw movie micrographs. A total of 2,094,437 particles were picked automatically. After 2D classification, a total of 1,789,831 particles were selected and used for 3D classification. Based on the quality of the four 3D classes, 590,043 particles were retained for further 3D reconstruction, refinement, and postprocessing, resulting in a 2.8 Å average resolution 3D map.

For the Dnf1–Lem3 structure in the E1 state, we collected 4074 raw movie micrographs. A total of 1,358,323 particles were picked automatically. After 2D classification, a total of 1,089,904 particles were selected and used for 3D classification. Based on the quality of the four 3D classes, 473,761 particles were selected for further 3D reconstruction, refinement, and postprocessing, resulting in a 3.1 Å average resolution 3D map.

For the Dnf1–Lem3 structure in the E1P-ADP state, we collected 6038 raw movie micrographs. A total of 2,257,199 particles were picked automatically. After 2D classification, a total of 1,789,831 particles were selected and used for 3D classification. Based on the quality of the four 3D classes, 716,510 particles were selected for further 3D reconstruction, refinement, and postprocessing, resulting in a 3.2 Å average resolution 3D map.

For the Dnf2–Lem3 structure in the E1 state, we collected 1438 raw movie micrographs. A total of 509,555 particles were picked automatically. After 2D classification, a total of 291,438 particles were selected and used for 3D classification. Based on the quality of the four 3D classes, 190,753 particles were retained for further 3D reconstruction, refinement, and postprocessing, resulting in a 3.1 Å average resolution 3D map.

For the Dnf2–Lem3 structure in the E1-AMPPCP state, we collected 3665 raw movie micrographs. A total of 844,464 particles were picked automatically. After 2D classification, a total of 519,537 particles were selected and used for 3D classification. Based on the quality of the four 3D classes, 315,846 particles were selected for further 3D reconstruction, refinement, and postprocessing, resulting in a 3.8 Å average resolution 3D map.

For the Dnf2–Lem3 structure in the E1P-ADP and E2P-transition states, we collected 5683 raw movie micrographs. A total of 1,010,708 particles were picked automatically. After 2D classification, a total of 593,683 particles were retained and used for 3D classification. Based on the conformation of the six 3D classes, the E1P-ADP and E2P-transition states were separated. For the E1P-ADP state, 266,218 particles were selected for further 3D reconstruction, refinement, and postprocessing, resulting in a 4.0 Å average resolution 3D map. The Dnf1 A domain was flexible and was not modeled. The N domain density was weak but clearly present. This domain was modeled as a rigid body and was excluded from final atomic model refinement. For the E2P-transition state, 127,442 particles were selected for further 3D reconstruction, refinement, and postprocessing, resulting in a 4.0 Å average resolution 3D map.

For the Dnf2–Lem3 in the E2P state, we collected 3260 raw movie micrographs. A total of 1,095,601 particles were picked automatically. After 2D classification, a total of 1,065,397 particles were retained and used for 3D classification. Based on the quality of the four 3D classes, 763,661 particles were selected for further 3D reconstruction, refinement, and postprocessing, resulting in a 3.5 Å average resolution 3D density map.

## Structural modeling, refinement, and validation

We first built the model of Dnf1–Lem3 in the E2P state at 2.8 Å resolution. We generated the initial model based on the structure of Drs2–Cdc50 (PDB ID 6PSY) by SWISSMODEL (https://swissmodel.expasy.org), and then manually corrected in COOT and Chimera (*Emsley et al., 2010*; *Pettersen et al., 2004*). The complete model of Dnf1–Lem3 was refined by real-space refinement in the PHENIX program and subsequently adjusted manually in COOT. Using the model of Dnf1–Lem3 in E2P as reference, models of Dnf1–Lem3 in the E1 and E1P-ADP states and of Dnf2–Lem3 in the E1, E1-ATP, E1P-ADP, E2P transition, and E2P states were built and refined using COOT, Chimera, and PHENIX. Finally, all models were validated using MolProbity in PHENIX (*Adams et al., 2010*; *Chen et al., 2010*). Structural figures were prepared in Chimera (*Pettersen et al., 2004*) and PyMOL (https://pymol.org/2/).

## Yeast strains and plasmid construction

Strains and plasmid used in the study are listed in *Supplementary files 2* and *3* (*Sikorski and Hieter, 1989*; *Hua et al., 2002*; *Hua and Graham, 2003*). All yeast culture reagents were purchased from Sigma-Aldrich and BD Scientific. Yeast strains were grown in YPD or minimal selective media. Yeast transformation was performed using the standard LiAC-PEG method. For plasmid shuffling assays, 10-fold serial dilutions starting from a cell suspension with cell density of $OD_{600}$ = 1 were spotted on synthetic complete media (SD) and SD-5-FOA and incubated at 30°C for 2 d before imaging. DNA constructs and mutations were created by Gibson assembly and quick-change mutagenesis.

## Yeast cell lysate preparation and immunoblotting

Yeast cell expressing 3X-FLAG tagged-Dnf1 and mutant strains were grown in selective minimal media to mid log phase. 2 OD units of cells were harvested and pellet was resuspended in 100 µl of SDS/urea sample buffer [40 mM Tris·HCl (pH 6.8), 8 M urea, 0.1 mM EDTA, 1% 2-mercaptoethanol (vol/vol), 5% SDS (wt/vol), and 0.25% bromophenol blue (wt/vol)] and equal volume of glass beads. Yeast cell lysate from 0.5 OD was separated on 4–20% gradient SDS–PAGE gel (Bio-Rad Cat. #4561095) and transferred to PVDF membranes. Dnf1 was probed using primary antibody anti-FLAG M2 antibody (Sigma–Aldrich) (1:2500) and secondary antibody Goat anti-Mouse HRP.

## Lipid uptake assay

All the lipids used in this study were purchased from Avanti Polar Lipids. NBD-lipid uptake was performed as previously described (*Roland et al., 2019*). Cells were grown to the mid-log phase with a cell density of approximately $OD_{600}$ = 1. Cells were collected for each strain. NBD-PC, NBD-PE, and NBD-GlcCer were dried and resuspended in 100% ethanol, then added to the designated ice-cold growth medium at a final concentration of 2 µg/mL. Cells were incubated on ice with the pre-chilled NBD-lipid+medium for 30 min. After the incubation, the cells were washed twice with pre-chilled SA medium (SD medium + 2% [w/v] sorbitol + 20 mM $NaN_3$) supplemented with 4% (w/v) fatty acid-free

BSA. Cells were resuspended in SA with 5 μM propidium iodide (PI) and NBD-lipid uptake was immediately measured by flow cytometry. The order in which the samples were prepared and processed was varied in each experiment to reduce potential experimental error. Each experiment consisted of three independent biological replicates for each strain, repeated three times ($n = 9$).

### Fluorescence microscopy

Cells were grown in minimal synthetic dropout medium to log phase. Cells were washed with fresh medium three times and resuspended in fresh SD medium. Cells were then mounted on glass slides and observed immediately at room temperature. Images were acquired using a DeltaVision Elite Imaging System (GE Healthcare Life Sciences, Pittsburgh, PA) equipped with a 100 × objective lens followed by deconvolution using softWoRx software (GE Healthcare Life Science).

Supplementary Information is available in the online version of the paper.

## Acknowledgements

Cryo-EM images were collected in the David Van Andel Advanced Cryo-Electron Microscopy Suite at Van Andel Institute. We thank Gongpu Zhao and Xing Meng for facilitating data collection and David Nadziejka for proofreading the manuscript. The Vanderbilt University Medical Center Flow Cytometry Shared Resource is supported by NIH grants to the Vanderbilt Ingram Cancer Center (P30-CA68485) and the Vanderbilt Digestive Disease Research Center (P30-DK058404). This work was supported by the U.S. National Institutes of Health (CA231466 to HL and GM107978 to TRG) and the Van Andel Institute (to HL).

## Additional information

### Funding

| Funder | Grant reference number | Author |
| --- | --- | --- |
| National Institutes of Health | CA231466 | Huilin Li |
| National Institutes of Health | GM107978 | Todd R Graham |
| Van Andel Research Institute | | Huilin Li |

The funders had no role in study design, data collection and interpretation, or the decision to submit the work for publication.

### Author contributions

Lin Bai, Conceptualization, Resources, Data curation, Formal analysis, Supervision, Validation, Investigation, Visualization, Writing - original draft, Project administration, Writing - review and editing; Qinglong You, Resources, Investigation, Writing - review and editing; Bhawik K Jain, Formal analysis, Investigation, Methodology, Writing - review and editing; H Diessel Duan, Investigation; Amanda Kovach, Investigation, Writing - review and editing; Todd R Graham, Conceptualization, Resources, Formal analysis, Supervision, Validation, Methodology, Writing - review and editing; Huilin Li, Conceptualization, Resources, Supervision, Funding acquisition, Validation, Project administration, Writing - review and editing

### Author ORCIDs

Bhawik K Jain (iD) http://orcid.org/0000-0002-1362-6139
Todd R Graham (iD) http://orcid.org/0000-0002-3256-2126
Huilin Li (iD) https://orcid.org/0000-0001-8085-8928

### Decision letter and Author response

Decision letter https://doi.org/10.7554/eLife.62163.sa1
Author response https://doi.org/10.7554/eLife.62163.sa2

# Additional files

## Supplementary files

- Supplementary file 1. Cryo-EM data collection, refinement, and validations.
- Supplementary file 2. List of plasmids used in the study.
- Supplementary file 3. List of strains used in the study (knockouts are *KanMX* replacements unless otherwise indicated).
- Transparent reporting form

## Data availability

The cryo-EM 3D maps and the corresponding atomic models of the Dnf1-Lem3 complex have been deposited at the EMDB database and the RCSB PDB with the respective accession codes of EMD-23069 and 7KY6 (apo E1), EMD-23074 and 7KYB (E1P-ADP), EMD-23077 and 7KYC (E2P). The cryo-EM 3D maps and the corresponding atomic models of the Dnf2-Lem3 complex have been deposited at the EMDB database and the RCSB PDB with the respective accession codes of EMD-23070 and 7KY7 (apo E1), EMD-23071 and 7KY8 (E1-ATP), EMD-23072 and 7KY9 (E1P-ADP), EMD-23068 and 7KY5 (E2P-transition), and EMD-23073 and 7KYA (E2P).

The following datasets were generated:

| Author(s) | Year | Dataset title | Dataset URL | Database and Identifier |
|---|---|---|---|---|
| Bai L, You Q, Jain BK, Duan HD, Kovach A, Graham TR, Li H | 2020 | Cryo-EM 3D map of Dnf1-Lem3 in the apo E1 state | https://www.ebi.ac.uk/pdbe/emdb/EMD-23069 | Electron Microscopy Data Bank, EMD-23069 |
| Bai L, You Q, Jain BK, Duan HD, Kovach A, Graham TR, Li H | 2020 | Atomic model of Dnf1-Lem3 in the apo E1 state | https://www.rcsb.org/structure/7KY6 | RCSB Protein Data Bank, 7KY6 |
| Bai L, You Q, Jain BK, Duan HD, Kovach A, Graham TG, Li H | 2020 | Cryo-EM 3D map of Dnf1-Lem3 in the E1P-ADP state | https://www.ebi.ac.uk/pdbe/emdb/EMD-23074 | Electron Microscopy Data Bank, EMD-23074 |
| Bai L, You Q, Jain BK, Duan HD, Kovach A, Grahan TR, Li H | 2020 | Atomic model of Dnf1-Lem3 in the E1P-ADP state | https://www.rcsb.org/structure/7KYB | RCSB Protein Data Bank, 7KYB |
| Bai L, You Q, Jain BK, Duan HD, Kovach A, Graham TR, Li H | 2020 | Cryo-EM 3D map of Dnf1-Lem3 in the E2P state | https://www.ebi.ac.uk/pdbe/emdb/EMD-23077 | Electron Microscopy Data Bank, EMD-23077 |
| Bai L, You Q, Jain BK, Duan HD, Kovach A, Graham TR, Li H | 2020 | Atomic model of Dnf1-Lem3 in the E2P state | https://www.rcsb.org/structure/7KYC | RCSB Protein Data Bank, 7KYC |
| Bai L, You Q, Jain BK, Duan HD, Kovach A, Graham TR, Li H | 2020 | Cryo-EM 3D map of Dnf2-Lem3 in the apo E1 state | https://www.ebi.ac.uk/pdbe/emdb/EMD-23070 | Electron Microscopy Data Bank, EMD-23070 |
| Bai L, You Q, Jain BK, Duan HD, Kovach A, Graham TR, Li H | 2020 | Atomic model of Dnf2-Lem3 in the apo E1 state | https://www.rcsb.org/structure/7KY7 | RCSB Protein Data Bank, 7KY7 |
| Bai L, You Q, Jain BK, Duan HD, Kovach A, Graham TR, Li H | 2020 | Cryo-EM 3D map of Dnf2-Lem3 in the E1-ATP state | https://www.ebi.ac.uk/pdbe/emdb/EMD-23071 | Electron Microscopy Data Bank, EMD-23071 |
| Bai L, You Q, Jain BK, Duan HD, Kovach A, Graham | 2020 | Atomic model of Dnf2-Lem2 in the E1-ATP state | https://www.rcsb.org/structure/7KY8 | RCSB Protein Data Bank, 7KY8 |

| TR, Li H | | | | | |
|---|---|---|---|---|---|
| Bai L, You Q, Jain BK, Duan HD, Kovach A, Graham TR, Li H | 2020 | Cryo-EM 3D map of Dnf2-Lem3 in the E1P-ADP state | https://www.ebi.ac.uk/pdbe/emdb/EMD-23072 | Electron Microscopy Data Bank, EMD-230 72 | |
| Bai L, You Q, Jain BK, Duan HD, Kovach A, Graham TR, Li H | 2020 | Atomic model of Dnf2-Lem3 in the E1P-ADP state | https://www.rcsb.org/structure/7KY9 | RCSB Protein Data Bank, 7KY9 | |
| Bai L, You Q, Jain BH, Duan HD, Kovach A, Graham TR, Li H | 2020 | Cryo-EM 3D map of Dnf2-Lem3 in the E2P-transition state | https://www.ebi.ac.uk/pdbe/emdb/EMD-23068 | Electron Microscopy Data Bank, EMD-230 68 | |
| Bai L, You Q, Jain BK, Duan HD, Kovach A, Graham TR, Li H | 2020 | Atomic model of Dnf2-Lem3 in the E2P transition state | https://www.rcsb.org/structure/7KY5 | RCSB Protein Data Bank, 7KY5 | |
| Bai L, You Q, Jain BK, Duan HD, Kovach A, Graham TR, Li H | 2020 | Cryo-EM 3D map of Dnf2-Lem3 in the E2P state | https://www.ebi.ac.uk/pdbe/emdb/EMD-23073 | Electron Microscopy Data Bank, EMD-230 73 | |
| Bai L, You Q, Jain BK, Duan HD, Kovach A, Graham TR, Li H | 2020 | Atomic model of Dnf2-Lem3 in the E2P state | https://www.rcsb.org/structure/7KYA | RCSB Protein Data Bank, 7KYA | |

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
