## [Decision Letter]

**Acceptance summary:**

Your study nicely shows that PC and PS flippases use a common mechanism for flipping these lipids between leaflets of biological membranes.

**Decision letter after peer review:**

Thank you for submitting your article "Transport mechanism of class-3 P4 ATPase lipid flippases" for consideration by *eLife*. Your article has been reviewed by three peer reviewers, including Christopher G Burd as the Reviewing Editor and Reviewer #1, and the evaluation has been overseen by Vivek Malhotra as the Senior Editor.

The reviewers have discussed the reviews with one another and the Reviewing Editor has drafted this decision to help you prepare a revised submission.

Summary:

The manuscript presents a structural analysis of the transport cycle of P4 ATPases, which mediate phospholipid translocation between leaflets of biological membranes. Eight structures of two yeast class-3 P4 ATPases, Dnf1- Lem3 and Dnf2-Lem3, in five transport states were determined using cryo-electron microscopy methods. Structure-based mutagenesis of mutant enzymes expressed in yeast cells were used to test and validate mechanistic predictions based on the structures. While the overall structures or Dnf1- Lem3 and Dnf2-Lem3 are quite similar to previously determined structures of these and other P4 ATPases, several new pieces of information were obtained. Most important is the identification of two lipid binding sites that are inferred to define the sites of substrate entry on the exofacial membrane leaflet and substrate exit on the cytoplasmic leaflet. A second major point is the inference of an unexpected role for the Lem3 β subunit in substrate recognition and inter-domain contacts that regulate transport activity. Finally, the identification of the conformational transitions during substrate flipping revealed that they are essentially identical to that of the class^-1^ P4 ATPases, therefore indicating that the pathways and mechanisms involved in lipid flipping are likely identical between enzymes in all classes.

Essential revisions:

1) Recent phylogenetic analysis revealed a new classification that groups Dnf1/2, ATP8A1 and Drs2 into the same class (Palmgren et al., 2019). This study should be cited in the manuscript and the revised classification considered in the interpretation of the results.

2) It is essential that the study provide evidence that the purified proteins under investigation display ATPase activity that can be stimulated by substrate lipids (but not by non-substrate lipids) and blocked by the ATP analogs/inhibitors. Published structural studies of relevant flippases (Bai et al., 2019, Timcenko et al., 2019, Hiraizumi et al., 2019) each include a demonstration the proteins under investigation are active and the reviewers consider this a standard that should be maintained. At a minimum, the authors should measure ATPase activity of the native sequence proteins in the presence/absence of ATP analogs/inhibitors.

3) The functional analyses of flippase mutants correlate changes in lipid uptake activity with specific mutations and attribute differences in lipid uptake activity to these mutations. The authors should establish if differences in expression levels of the mutant proteins can account (or not) for the observed changes in lipid uptake activity. A standard approach is to compare protein levels in cell extracts by immune-blotting. Such an analysis (or equivalent) should be included, if not for all the proteins tested, at least the ones that are most relevant to the authors conclusions.

---

## [Author Response]

Essential revisions:1) Recent phylogenetic analysis revealed a new classification that groups Dnf1/2, ATP8A1 and Drs2 into the same class (Palmgren et al., 2019). This study should be cited in the manuscript and the revised classification considered in the interpretation of the results.

In light of this recommendation, we have de-emphasized the phylogenetic grouping throughout the manuscript and placed greater emphasis on the differences in substrate specificity for Dnf1/2-Lem3 relative to their structurally defined homologs. We have included the Palmgren citation within the Introduction and described this work in the Discussion section as it pertains to our results:

“In a recent phylogenetic classification, P4-ATPases are divided into P4A, P4B and P4C subclasses (Palmgren et al., 2019). […] While structural knowledge obtained thus far raises the possibility that all P4-ATPases employ a similar substrate-transport mechanism, more definitive support awaits further structure and functional analyses of P4-ATPases in the P4B and P4C classes”.

2) It is essential that the study provide evidence that the purified proteins under investigation display ATPase activity that can be stimulated by substrate lipids (but not by non-substrate lipids) and blocked by the ATP analogs/inhibitors. Published structural studies of relevant flippases (Bai et al., 2019, Timcenko et al., 2019, Hiraizumi et al., 2019) each include a demonstration the proteins under investigation are active and the reviewers consider this a standard that should be maintained. At a minimum, the authors should measure ATPase activity of the native sequence proteins in the presence/absence of ATP analogs/inhibitors.

We have re-purified Dnf1 and Dnf2 enzymes and performed ATP hydrolysis assay for both Dnf1 and Dnf1 in the presence of a substrate (glucosylceramide; GlcCer) and a non-substrate (sphingomyelin; SM) (new Figure 1C). We also demonstrate the concentration dependent stimulation of the ATPase activity of both Dnf1 and Dnf2 by substrate (new Figure 1D).

3) The functional analyses of flippase mutants correlate changes in lipid uptake activity with specific mutations and attribute differences in lipid uptake activity to these mutations. The authors should establish if differences in expression levels of the mutant proteins can account (or not) for the observed changes in lipid uptake activity. A standard approach is to compare protein levels in cell extracts by immune-blotting. Such an analysis (or equivalent) should be included, if not for all the proteins tested, at least the ones that are most relevant to the authors conclusions.

We have now performed control experiments to demonstrate that the protein levels are not affected by any of the introduced mutations (Figure 4—figure supplement 1).